# Accurate Prediction of Knee Angles during Open-Chain Rehabilitation Exercises Using a Wearable Array of Nanocomposite Stretch Sensors

**DOI:** 10.3390/s22072499

**Published:** 2022-03-24

**Authors:** David S. Wood, Kurt Jensen, Allison Crane, Hyunwook Lee, Hayden Dennis, Joshua Gladwell, Anne Shurtz, David T. Fullwood, Matthew K. Seeley, Ulrike H. Mitchell, William F. Christensen, Anton E. Bowden

**Affiliations:** 1Department of Mechanical Engineering, Brigham Young University, Provo, UT 84602, USA; dswood@sandia.gov (D.S.W.); kurtjen1995@gmail.com (K.J.); acranetg@gmail.com (A.C.); dfullwood@byu.edu (D.T.F.); 2Department of Exercise Science, Brigham Young University, Provo, UT 84602, USA; hyunwook.lee31@gmail.com (H.L.); haydo1214@gmail.com (H.D.); matt_seeley@byu.edu (M.K.S.); rike_mitchell@byu.edu (U.H.M.); 3Department of Statistics, Brigham Young University, Provo, UT 84602, USA; joshegladwell@gmail.com (J.G.); annekfagerburg@gmail.com (A.S.); william@stat.byu.edu (W.F.C.)

**Keywords:** nanocomposite stretch sensors, smart textile, rehabilitation

## Abstract

In this work, a knee sleeve is presented for application in physical therapy applications relating to knee rehabilitation. The device is instrumented with sixteen piezoresistive sensors to measure knee angles during exercise, and can support at-home rehabilitation methods. The development of the device is presented. Testing was performed on eighteen subjects, and knee angles were predicted using a machine learning regressor. Subject-specific and device-specific models are analyzed and presented. Subject-specific models average root mean square errors of 7.6 and 1.8 degrees for flexion/extension and internal/external rotation, respectively. Device-specific models average root mean square errors of 12.6 and 3.5 degrees for flexion/extension and internal/external rotation, respectively. The device presented in this work proved to be a repeatable, reusable, low-cost device that can adequately model the knee’s flexion/extension and internal/external rotation angles for rehabilitation purposes.

## 1. Introduction

The rate of invasive knee surgeries is rising. The number of total knee arthroplasty (TKA) procedures has steadily increased over the past three decades [1,2]. Future projections estimate that the number will increase to 1.16–3.48 million annually by 2030 [2,3], resulting in about 90 procedures per orthopedic surgeon per year in the United States [4]. ACL reconstructions are also expected to increase [5]. There is a proportional need for post-operative physical therapy. However, recent estimates show a decreasing ratio of physical therapists (PTs) per unit of population [6,7], partly due to burnout and career dissatisfaction [8] among PTs. The imbalance between the rising number of patients requiring rehabilitation and a decreasing workforce is expected to grow.

Home-based rehabilitation methods have been developed to accommodate recent trends relating to available PTs and the number of patients requiring therapy. Compared to inpatient or outpatient rehabilitation, home-based interventions are less expensive [9,10,11,12] and more accessible [12,13]. One critique of at-home rehabilitation, however, is decreased regimen adherence that is usually seen in unguided or unsupervised rehabilitation [14]. However, important advantages of at-home rehabilitation, as described by patients, include ease of use of many currently available systems, decreased stress, and “not having to travel when we are in pain” [13].

One solution to increase rehabilitation availability and patient accountability is sensor-based technology. Such technology could potentially maintain quality of rehabilitative care despite increasing patient-to-PT ratios. Sensor-based technology could also give the PT quantitative feedback on the patient’s progress while performing at-home exercises. However, sensor-based rehabilitation also has drawbacks: battery limitations, sensor drift over time [15], device cost, and mobility. Furthermore, depending on the specific approach, and its implementation, limitations might outweigh benefit and restrict widespread use of sensor technology in physical rehabilitation. For example, if a sensor-based device exhibits a large amount of drift, the error contained within the data may be large enough to render the system unusable. This work presents a cost-efficient, low-power device to facilitate passively supervised at-home rehabilitation via a knee sleeve instrumented with an array of piezoresistive sensors.

High-deflection strain gauges are desirable for biomechanical measurement applications because of their large strain capacity and sensitivity to changes in strain [16,17]. There are many types of high-deflection strain gauges used in biomechanics including liquid metal [18,19,20], gel-based [21,22], polymer optical fiber [23,24,25], and piezoresistive sensors. The most common filler materials used in high-deflection strain gauges are derivatives of carbon [26,27,28,29,30], due to the favorable mechanical properties, high electrical conductivity, and high thermal conductivity of carbon-based polymers.

A notable biomechanical application of polymer-based high-deflection strain gauges is smart textiles, garments ranging from braces to shirts that are instrumented with sensors to measure forces, positions, or pressure in the body [31]. For example, Shyr et al. [32] estimated knee and elbow angles using a woven elastic textile. Gholami et al. [33] developed a pair of leg tights capable of measuring knee flexion and extension with a high degree of repeatability via machine learning regressors. These and other examples demonstrate the potential for wearable sensors in TKA rehabilitation.

Although knee flexion and extension are essential for successful TKA rehabilitation, the knee should not be considered a single degree of freedom joint. It has been well established that transverse plane (internal/external of the tibia) rotation occurs naturally in gait [34] and is a conjunct (or linked) movement necessary for full range of motion [35]. All mentions of internal/external in this work are concerning the tibia unless otherwise noted. Post-operative tibia rotation correlates with post-operative flexion/extension [36], but may not necessarily be correctly rehabilitated due to prosthesis misalignment [37] or alterations in the patient’s gait patterns [36]. Recovery of normal ranges of rotation of the tibia is especially imperative for flexion while bearing weight [38]. Inadequate internal rotation of the tibia is a common cause of chronic pain after TKA [37]. Thus, for a complete recovery of the knee’s pre-operative range of motion with decreased pain, flexion/extension and internal/external rotation should be simultaneously monitored and improved during rehabilitation.

This work utilizes a triphasic silicone/nickel composite [39,40,41,42,43,44,45,46,47,48,49,50,51,52,53,54,55] with two different nickel filler materials—nickel nanostrands (NiNs, (18.76 wt%); Figure 1a) and nickel-coated carbon fiber (NCCF, (3.13 wt%), Figure 1b). The resulting sensors are inversely piezoresistive (negative correlation between strain and electrical resistance). A single triphasic silicone/nickel sensor was previously used to measure knee flexion/extension [54], but an array of these sensors can measure additional degrees of freedom in the knee. We hypothesize that, by placing piezoresistive strain gauges in a garment across locations of the anterior knee, where the skin experiences the greatest variance in strain, we can accurately model and predict two rotational degrees of freedom: flexion/extension and tibia/femur rotation, during open-chain knee flexion.

The device presented in this work was used to measure two rotational degrees of freedom (flexion/extension and internal/external rotation) of the knee in healthy subjects to determine the accuracy and repeatability of a 16-sensor, skin-mounted instrumented sleeve. Future application is intended to occur in physical therapy applications during TKA rehabilitation. The technology represents a cost-efficient instrument that can eventually be used to enhance at-home rehabilitation for TKA patients by providing a quantifiable measurement of two rotational degrees of freedom. Future work will incorporate these measurements into a real-time biofeedback system for improving the quality of at-home physical therapy exercises [56].

## 2. Materials and Methods

### 2.1. Data Collection

The sensor system for this work, shown in Figure 2, was mounted onto an off-the-shelf 88% copper nylon, 12% spandex knee sleeve fitted so as not to squeeze the knee joint, yet tight enough to not migrate during exercise. The positioning of an array of piezoresistive sensors was developed based on previous work [57]. The fitting of the device was devised with the guidance of an orthopedic surgeon and experienced PT. The instrumented knee sleeve conforms to the skin of the knee without adhering to the skin and can be directly donned and doffed by the user. A custom printed circuit board was developed to read and simultaneously log data from all sixteen sensors. The sensors were connected with a multiplexer into a Wheatstone bridge circuit that was excited with a 5-Volt square wave at 300 Hz. The printed circuit board was externally powered and logged the data to an accompanying computer through a serial connection.

The sensor component materials, nickel nanostrands (18.76 wt%), nickel-coated carbon fibers (3.13 wt%), and Ecoflex 00-30 silicone (Smooth-On Inc., Macungie, PA, USA), were mixed and the resultant slurry was pressed into aluminum molds of three different lengths. Five sensors were 21.0 mm long, ten were 31.2 mm, and one was 42.0 mm in length. All sensors were 0.8 mm thick and 5.0 mm wide. Copper wires were embedded into the ends of every sensor by placing 32-gauge wires within the slurry material before curing at 190 °F for 90 min. Then, a thin layer of Libra gloss silicone from Zodiac (Kennesaw, GA, USA) was spread onto a nylon/spandex knee sleeve over the desired location for each sensor and cured at 190 °F for 30 min. This silicone layer created a base under each sensor that prevented the Ecoflex silicone sensor adhesive from wicking into the nylon during the curing process. Finally, sixteen sensors were adhered to the silicone footings and cured at 190 °F for 30 min. To connect the sensors to the circuit board described previously, 36-gauge copper magnet wire was embroidered into a separate piece of rayon spandex cloth and secured to the sleeve with stitch points between the sensors. By ensuring the additional material fit more loosely than the knee sleeve, the layer of wires did not inhibit the motion of the knee sleeve on a subject’s skin. The approximate manufacturing cost of this sensing system was $7–8 plus electronics (roughly $20) and the cost of an off-the-shelf knee sleeve.

Previous work identified positions around the knee that are sensitive to knee motion and informed the layout of the sensors in the device presented here [57]. The sensor array was designed to capture redundant information in the array due to overlaps in sensor orientation and positioning. Because PT goals are oriented strictly around knee flexion/extension and internal/external rotation, the present work only included these movements, but future work may include knee translation, valgus/varus motion, and patella movement.

To develop a predictive model for the sensor array and validate the array performance, an optical tracking system was used in conjunction with the sensor array. Twenty one markers were placed on the right leg and pelvis to measure knee kinematics (Figure 3). Motion capture data were collected at 100 Hz, and sensor data were captured at 18.75 Hz/sensor. The same computer captured both sets of data so the two sets could be synchronized with respect to time using the computer’s local timestamp. Knee angles were calculated from the marker data using a generic lower-body model in Visual3D (C-Motion, Germantown, MD).

Eighteen healthy subjects (9 male, 9 female) between the ages of 20 and 47 (27.4 ± 7.1 years, mean ± SD), of varying body sizes (body mass indexes ranged from 18.3 to 33.9, with a mean of 23.9 ± 4.1) participated in this study, to determine the accuracy and repeatability of the multi-sensor system. Information about the cohort can be found in Table 1. Exclusion criteria for these subjects were past knee or hip surgeries or major non-surgical knee injuries within the past two years. Subjects performed open-chain knee flexion (OCKF), an exercise that replicates the range of motion of movements seen during post-operative TKA rehabilitation. OCKF was continually performed at an instructed pace of “two seconds up and two seconds down” for 60 s.

### 2.2. Modeling and Predictions

The collected data were used to produce subject-specific models and predict knee angles. A subject-specific model is defined in this work as a model that was trained on the data from a single subject and predicts knee angles of the same subject. Each model was trained and tested with ten-fold cross-validation from the data collected from one subject in the previous section. First, high-frequency noise was removed from the data using a Butterworth filter, with cutoff frequencies of 10 Hz for the marker motion data and 2 Hz for the strain sensor data. Kinematic experimental data are commonly filtered with a Butterworth filter with a cut-off frequency between 3 and 10 Hz [58]. The lower cut-off frequency of 2 Hz was used to filter the strain sensor data because the signal-to-noise ratio of the instrumented knee sleeve was higher than the motion capture data. A cut-off frequency that was double the frequency of the captured motion was used. Although it is lower than common cut-off frequencies, 2 Hz was experimentally found to provide the greatest increase in the strain sensor data signal-to-noise ratio. Once filtered, the motion capture data were downsampled to match the capture rate of the instrumented knee sleeve.

Unfiltered data from the sixteen sensors contained 1- and 2-point spikes that were more than triple the magnitude of the rest of the signal. Static electricity between the nylon–spandex knee sleeve material and exposed soldered points around the sensors likely caused these transient spikes. After these outlying spikes were eliminated from the sensor data using a Hampel filter (threshold = 2 SD), a Savitzky-Golay smoothing filter using a centered window smoothed the sensor data. At this point, 16 additional variables were created from the derivatives of each sensor’s data with time. Derivatives were calculated using finite differences between consecutive data points. Lastly, the mean values were subtracted from the sensor and derivative signals, and the data were scaled to unit variance before predicting knee angles. The model’s explanatory variables were the scaled sensor outputs and their derivatives. Figure 4 shows the post-processing pipeline.

Eighteen machine learning models were evaluated as candidates for mapping the composite sensor responses to the knee angles. These models include linear regression, neural networks, random forest regressors, adaptive boosted random forests, etc. All included models are listed in Table A1 and Table A2. The root mean square error (RMSE) and coefficient of determination (R^2^) were used to assess model accuracy. The model with the lowest mean RMSE over all subjects was selected, excluding models with excessively large outlier RMSEs. Removing models with outlying RMSEs decreased the likelihood of the chosen model failing for out-of-sample participants. To further test the validity of the instrumented knee sleeve, an analysis of bias and variability in predictions was performed to evaluate the predictive model’s agreement with the knee angles calculated from motion capture data.

Predictive modeling of data were completed using scikit-learn [59]. The results from ten separate models trained on different portions of a subject’s data were averaged to determine the robustness of the model’s predictive capability. Each model was trained on 90% of a subject’s data and tested on the remaining 10% of data. The training and testing data were separated in a way to preserve the time series component in the data. Multivariate machine learning regressor hyperparameters were optimized using the entire dataset from a subject using grid search and multivariate Bayesian optimization approaches. The average RMSE and R^2^ values from the ten unique models trained and tested on different portions of a subject’s data determined each model’s predictive power. Three instrumented knee sleeves were tested; therefore, both subject-specific and device-specific models were analyzed.

## 3. Results

### 3.1. Data Collection

Subjects performed exercises in similar ranges of motion with respect to knee angle; a complete record of knee angle ranges is found in Figure 5. Walking trials included knee flexion angles between $-2.9\pm 3.9^{\circ }$ and $58.7\pm 5.1^{\circ }$ (mean ± standard deviation), and OCKF trials included knee flexion angles between $-2.0\pm 4.8^{\circ }$ and $97.5\pm 12.2^{\circ }$. Walking trials included internal/external rotation angles between $-10.6\pm 7^{\circ }$ and $6.9\pm 12.7^{\circ }$, and OCKF trials included internal/external rotation angles between $6.5\pm 8.3^{\circ }$ and $9.1\pm 8.3^{\circ }$.

Data from motion capture indicated ranges of knee flexion/extension angles between $-2.0\pm 4.8^{\circ }$ and $97.5\pm 12.2^{\circ }$ and internal/external rotation angles between $6.5\pm 8.3^{\circ }$ and $9.1\pm 8.3^{\circ }$. All participants exhibited similar ranges of motion in flexion/extension and internal/external rotation and were within normal ranges of motion (see Figure 5). Participants completed an average of 14 movements in 60 s.

### 3.2. Modeling and Predictions

Figure 6 shows representative sensor data and its calculated derivative that was used as explanatory variables in the predictive models. Table A1 and Table A2 summarize the results from all 18 models. Adaptive boosting of a random forest regressor (RFR) predicted flexion/extension angles and internal/external rotation most accurately. The accuracy of the model’s predictions is indicated by both the R^2^ and RMSE values. Figure 7 shows the performance of each model in the two rotational degrees of freedom of interest. The average coefficient of determination (R^2^) of flexion/extension and internal/external rotation for the final subject-specific models were 0.940 and 0.731, respectively. The average RMSE across all subjects was 7.6° for flexion/extension and 1.8° for internal/external rotation. By including the derivatives of the sensor outputs in the model, the average RMSE improvement in the flexion/extension and internal/external rotation degrees of freedom were −16.9% and −9.3%, respectively. The average improvement of the R^2^ values in the flexion/extension and internal/external rotation degrees of freedom were 1.7% and 24.7%, respectively. All reported values are from models that included the signal’s derivative. Table 2 shows the accuracy of the device-specific models.

Figure 8 shows the dependency of the model’s bias and standard deviation as a function of angle of flexion/extension (Figure 8a,b) and internal/external rotation (Figure 8c,d). Figure 8 demonstrate that there is the most bias at the extreme of the model’s flexion/extension angle, yet the smallest standard deviation. The same holds for the model’s internal/external rotation. The largest standard deviation occurs in the middle of the range of knee angles (for both degrees of freedom), where the average bias is the smallest.

Using the presented pipeline, shown graphically in Figure 4, the resulting predictions and actual knee angles in Figure 7 were calculated to create the subject-specific models. The models correctly predicted the frequency of a motion and reasonably captured the magnitude of the extrema during OCKF. However, the magnitude of the models’ underestimation at the extrema is consistent (as shown in the bias plots in Figure 8a,c). Furthermore, the predictive model’s largest errors occurred at angles less than or equal to 0° flexion (with a straightened knee). This agrees with the bias results for all subjects shown in Figure 8.

The residuals of the predictive models’ bias showed a strong linear correlation with the actual knee angle, see Figure 8. Adding a linear regression model of the residuals to correct the bias was analyzed but proved to be statistically insignificant and was not included in these results. A paired t-test, between the models, including the linear regression of the residuals’ bias and those without, resulted in p-values greater than or equal to 0.191 for the R^2^ values and RMSE of both knee angles.

## 4. Discussion

An instrumented knee sleeve was developed, presented, and modeled using an adaptively boosted RFR model. The presented knee sleeve is a low-cost alternative to measuring knee kinematics outside a laboratory setting with comparable accuracy to current technologies. Unfortunately, during testing on two subjects, connectivity was lost between the sensors and the printed circuit board due to severed wires. Thus, the results from these two subjects resulted in failed models and were excluded from the predictive model analysis.

The predictive models’ most significant bias occurred at angles less than or equal to 0° flexion (an extended knee). This can be partially due to the laxity in the knee sleeve once the garment began to slip. After full flexion, the sleeve exhibited some buckling when the leg extended again. This buckling also occurred in several strain gauges near the patella and the insertion of the vastus lateralis, likely leading the model to believe the knee was in flexion (when in actuality, it was hyperextended), thus underestimating the peaks of the knee angle. The predictive models exhibited the most significant deviation in the middle of the range of motion. This is most likely caused by the high angular velocities at these angles and related to the inherent hysteresis caused by the polymer-based piezoresistive sensors’ viscoelastic nature. This has been seen previously in [54,60], and an attempt to compensate for this effect post hoc can be seen in [23]. Similar to all polymer-based wearable sensors, the sensors used in the current work exhibited viscoelastic effects such as creep, drift, and hysteresis. The magnitude of these effects was not found to affect the accuracy of the current analysis, and additional work characterizing these important sensor properties is currently underway. The impact of these errors is similar to previous IMUs [61,62,63,64,65] and high deflection strain gauges [54,66,67] used to measure joint angles. By including the derivatives of the sensor outputs in the model, the average improvement of the RMSE in the flexion/extension and internal/external rotation degrees of freedom were −16.9% and −9.3%, respectively. The average increase of the R^2^ values in the flexion/extension and internal/external rotation degrees of freedom were 1.7% and 24.7%, respectively. Accounting for differences in angular velocities significantly decreased the RMSE of the models developed in this work; accounting for angular velocities in optical fiber strain gauges results in similar improvements [68]. All results from the predictive models are for within-subject predictions, and the prediction of angles for previously unmeasured subjects would still have to be addressed.

Numerous studies have looked at methods to maximize the efficacy of at-home rehabilitation with biofeedback systems, namely teleconferencing with a PT [69,70,71,72], inertial measurement units (IMUs) [73,74], Nintendo Wii based systems/virtual reality (VR) [75,76,77,78], or PT guidance with robotic assistance [79]. Lockdowns associated with the COVID-19 pandemic have accelerated the development of devices for at-home rehabilitation [80]. However, telerehabilitation—rehabilitation that is still under the active guidance of a PT via video or telephone conferencing—continues to be the norm [15,69,71,72,81]. Telerehabilitation does little to relieve the increasing demand for TKA rehabilitation due to its reliance on one-to-one patient-to-therapist structure. Affordable IMUs are portable solutions that work for brief periods. However, issues with sensor drift [82] and calibration sensitivity [83] keep this solution from being widely adopted. Nintendo Wii-based systems are another promising solution. However, Nintendo Wii and VR-based systems address only stationary rehabilitation exercises, namely lateral weight shifting, multidirectional balance, and postural control [84]. Lastly, robotic assistance devices are the most accurate of these alternatives. However, they are also the most costly. Most robotic assistance measures are employed for inpatient use (as in [79]) but may be engineered for at-home use in the future. The sensor system presented in this work may be integrated into the current technologies to enhance their effectiveness or address their shortcomings. For example, the presented device may enhance telerehabilitation by relieving the workload of PTs. By combining the two approaches, most rehabilitation may be completed without direct PT observance. The presented device also allows the rehabilitation of many patients simultaneously, and a PT may intervene when necessary. This approach can increase the patient-to-therapist ratio without unduly increasing a PTs workload, yet still provide quantifiable feedback on a patient’s progress.

As noted in the Results section, our device could not fully capture the magnitude of the peaks and valleys in the flexion/extension or internal/external rotation degrees of freedom for all knee flexions for all subjects. Similar to other polymer-based devices [24], our device underestimated the extremes. The KneeHapp device developed by Haladjian et al. in [85] is a similar device that exhibited errors of similar magnitude to this work (4.82 ± 3.92°) for the flexion/extension angle. Other technologies, namely IMU-based devices (such as the one seen in [65]), overestimate the extremes. Additional filtering techniques are out of the scope of this work. However, they could be used in future works to mitigate this error since it was fairly consistent for each subject; i.e., the amount of underestimation was of a similar proportion to the actual peak angle for each knee flexion performed within each subject. The device presented in this work represents a cost-efficient instrument that can enhance at-home rehabilitation for TKA patients by providing a quantifiable measurement of two rotational degrees of freedom.

The present work focuses on the development and subject-specific modeling. However, the longer-term goals of the project are to generate device-specific models, and eventually a universal model that can be applied to all subjects. These goals will likely require a more inclusive incorporation of the viscoelastic characteristics of the sensors, as well as the inclusion of subject-specific demographic and anthropometric information.

We do note three limitations to this study. First, motion capture data were downsampled to equal the sampling rate of the instrumented knee sleeve resulting in some interpolation. Although data were interpolated with a quadratic fit with continuous derivatives, this step may have caused minute errors in the flexion/extension or internal/external rotation angles referenced as ground truth throughout the data collection. Second, the presented device was not adhered to the skin’s surface. Therefore, motion of the device relative to the skin’s surface within the data are possible and may add errors to the data if the device was not positioned correctly or was loosely fitted. This may be especially relevant to closed chain movement where the potential for movement of the device is amplified. Similar limitations have also been observed previously by Gholami et al. [33]. Third, there is debate around the accuracy of motion capture methods when measuring tibia internal/external rotation. Merriaux et al. found the three-dimensional positioning error of a motion capture system to be 2 mm during dynamic motion [86]. When using motion capture to measure internal/external rotation, Keizer and Otten concluded that rotations less than 1.70 degrees should be taken with caution due to the sum of errors with the camera system [87]. Additional errors can be introduced from improper marker placement on a limb’s anatomical landmarks, as well as from skin artifacts [88,89]. To minimize errors related to improper placement or motion relative to the skin, the same researcher placed markers on all participants to eliminate inter-researcher placement error. The researcher also used double-sided adhesive to attach the markers to all participants. Other techniques to more confidently measure bone motion may be used to mitigate this potential source of error. However, the sensor prediction data shown here reflected the motion capture data, despite the debate of the accuracy of the technique used.

In conclusion, the instrumented knee sleeve presented in this work accurately predicted two rotational degrees of freedom within the knee during OCKF. The viscoelastic nature of the piezoresistive sensors used to predict the knee angle was apparent and a source of error in predicting two rotational degrees of freedom at high angular velocities. Nevertheless, the instrumented knee sleeve presented in this work proved to be a repeatable, reusable, low-cost device that can adequately model the flexion/extension and internal/external rotation angles in the knee.

## Figures and Tables

**Figure 1 sensors-22-02499-f001:**
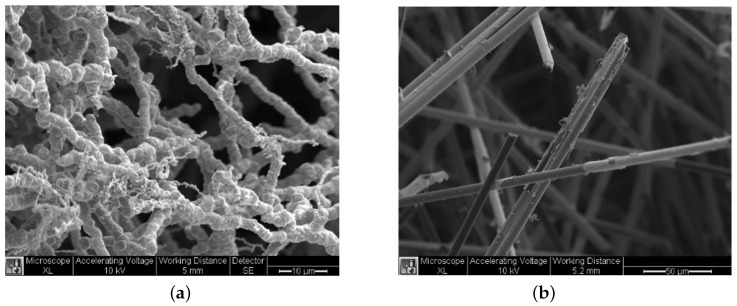
SEM images of the two electrically conductive filler materials. (**a**) SEM of NiNs. (**b**) SEM of NCCFs.

**Figure 2 sensors-22-02499-f002:**
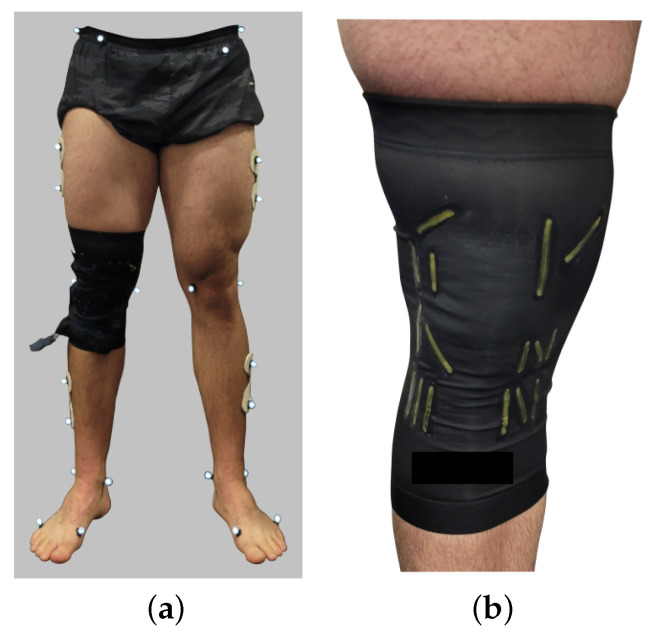
(**a**) shows the placement of the instrumented knee sleeve and markers used in the data collection portion of the study. The placement of 16 sensors within the knee sleeve are shown in yellow in (**b**). (**a**) Marker/sleeve placement. (**b**) Sensor placement.

**Figure 3 sensors-22-02499-f003:**
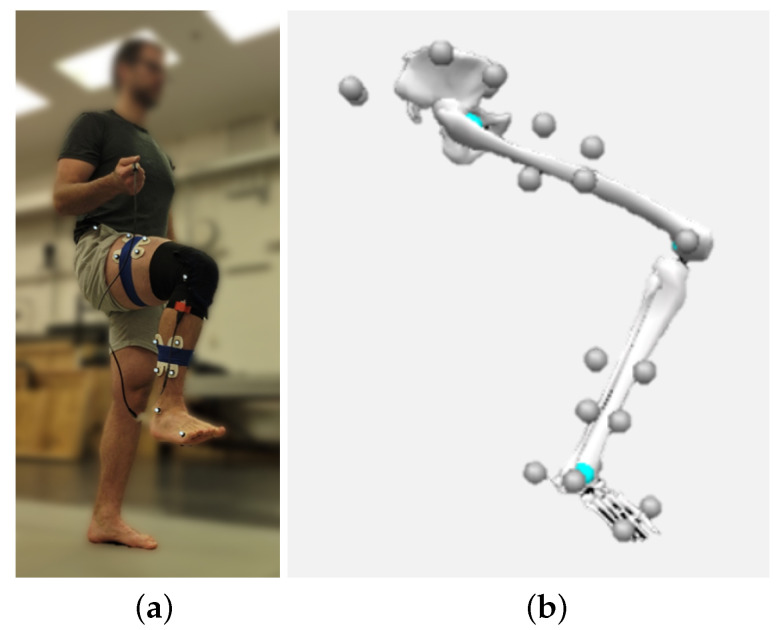
(**a**) Demonstration of open-chain knee flexion. (**b**) The captured motion with marker placement.

**Figure 4 sensors-22-02499-f004:**
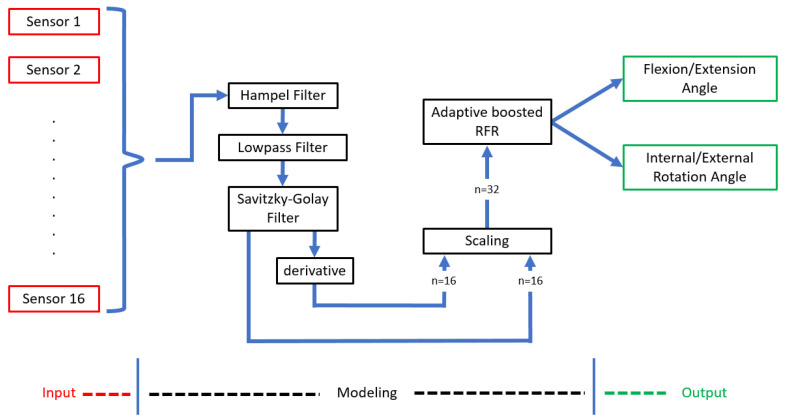
Flow chart of the pipeline used to predict knee angles.

**Figure 5 sensors-22-02499-f005:**
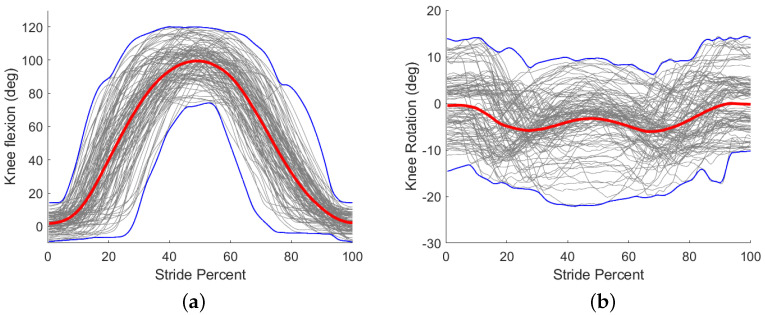
Knee angle ranges of all subjects of two rotational degrees of freedom. The red line represents the mean of all subjects. The blue lines represent the extrema. Positive angles correspond to knee flexion and internal rotation of the tibia. (**a**) Flexion/extension angle. (**b**) Tibia internal/external rotation.

**Figure 6 sensors-22-02499-f006:**
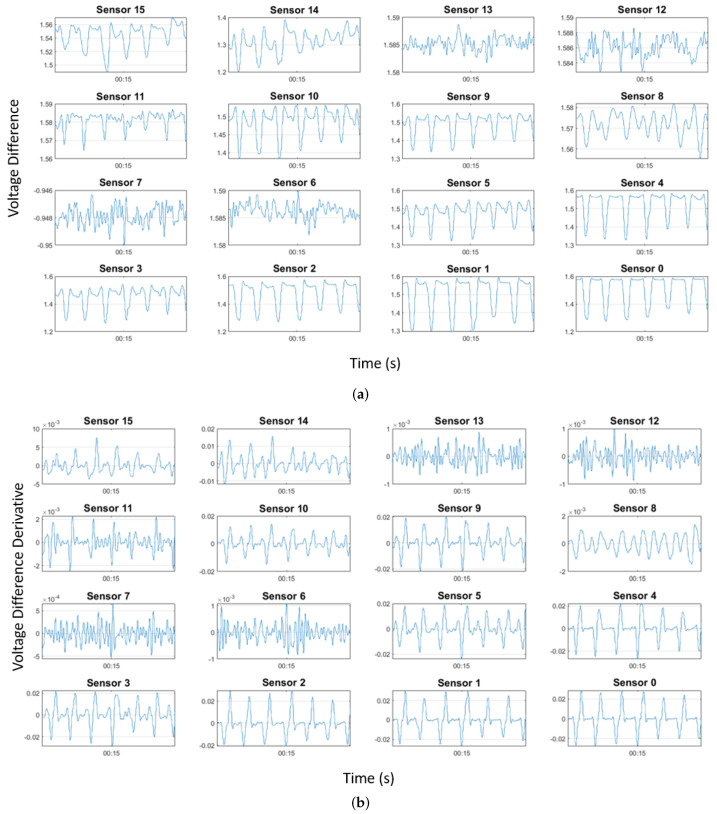
Representative sensor data and its derivative from 30 s of open chain knee flexion.

**Figure 7 sensors-22-02499-f007:**
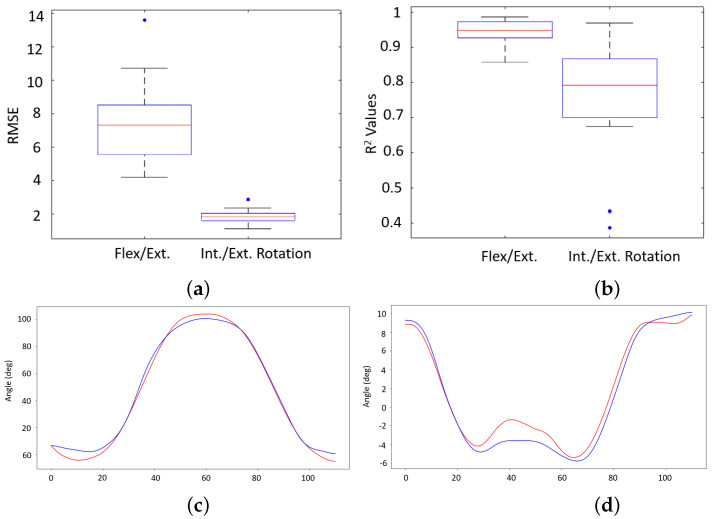
(**a**,**b**) show the distribution of the best model’s RMSE and R^2^ value in both degrees of freedom for subject-specific models. Red bars denote the overall average and whiskers denote standard deviations of the dataset. Samples of the models’ outputs (red) and actual knee angles (blue) are also shown. Positive angles correspond to knee flexion and internal rotation. (**c**,**d**) show representative data from one trial. (**a**) RMSE. (**b**) R^2^. (**c**) Predicted (red) versus actual (blue) flexion/extension knee angle. (**d**) Predicted (red) versus actual (blue)internal/external rotation angle.

**Figure 8 sensors-22-02499-f008:**
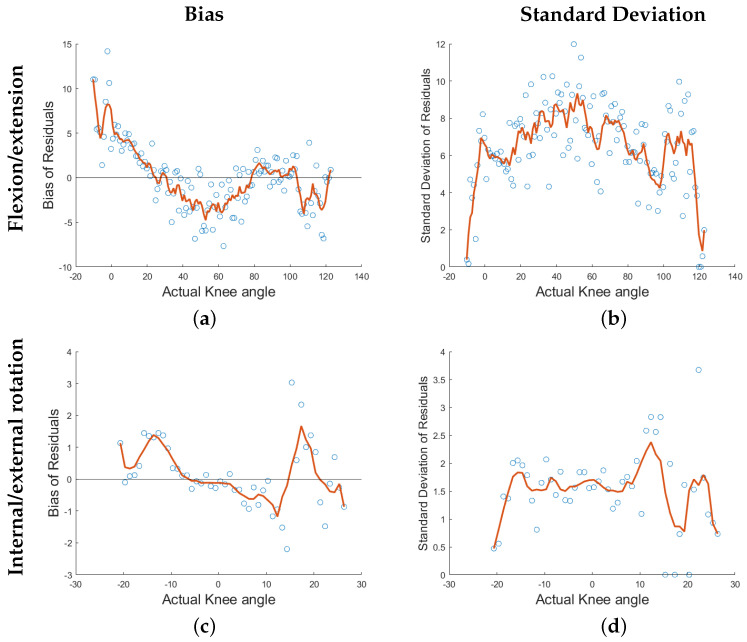
Distributions of all subjects’ bias and standard deviations in both degrees of freedom. Blue dots represent the local average across all subjects within 2 degree ranges. Lines represent the average of all subjects.

**Table 1 sensors-22-02499-t001:** Biometric information of the cohort of subjects.

Gender	Age	Height (cm)	Weight (kg)	BMI
M	30	185	93	27.2
M	29	189	101.7	28.5
M	33	179.5	87.6	27.2
M	29	181	74.3	22.7
M	21	183	72.8	21.7
M	47	179	71.3	22.3
F	22	161.5	58.4	22.4
M	24	182.5	65.9	19.8
F	40	174	74.8	24.7
F	25	180	77.8	24.0
F	24	165	54.2	19.9
F	21	168	60	21.3
F	21	168	59.1	20.9
F	26	162.75	48.5	18.3
M	20	187.5	71.3	20.3
F	31	163.5	68.8	25.7
M	28	177	93.8	29.9
F	23	166	93.5	33.9

**Table 2 sensors-22-02499-t002:** RMSE and $R^{2}$ values of the device-specific models.

Device	Flexion/Extension		Int./Ext. Rotation		Number of Participants
	RMSE	$R^{2}$	RMSE	$R^{2}$	
A	18.525	0.723	5.528	0.271	14
B	5.744	0.975	2.714	0.862	3
C	13.577	0.859	2.285	0.676	1

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
