# Peer review of "Accurate Prediction of Knee Angles during Open-Chain Rehabilitation Exercises Using a Wearable Array of Nanocomposite Stretch Sensors"

_sensors, 2022, doi:10.3390/s22072499_

Round 1
Reviewer 1 Report
In this manuscript, a knee sleeve instrumented with 16 inverse piezoresistive sensors for the purpose of predicting knee angles (both flexion/extension and internal/external angles) is presented. ML algorithms are used to generate the models matching the sensor recordings and the knee angles derived from motion capture systems. Comments are:
- The work in this manuscript can be considered as a modelling for the prediction of knee angles by using inverse piezoresistive sensors. In the context of TKA rehab, it is not clear how to obtain such a model when facing such a patient. For example, how to collect the sensor data and the motion capture data. Or you are going to generate a universal model for all the patients?
- It is not clear what the subject-specific models mean in this work. A further explanation is needed.
- “The sensor system for this work was mounted onto an off-the-shelf 88% copper nylon- 12% spandex knee sleeve that is fitted so as not to squeeze an anterior TKA incision yet tight enough to not migrate during exercise.” – it is not very clear to me what do you mean by not to squeeze an anterior TKA incision if all your subjects are healthy people.
- There is nowhere to find reference [55].
- “First, high-frequency noise were removed using a Butterworth filter, with cutoff frequencies of 10 Hz for the marker motion data and 2 Hz for the strain sensor data. Once filtered, the motion capture data were downsampled to match the capture rate of the instrumented knee sleeve.” – What is the rationale for choosing your cutoff frequencies? A further explanation is needed how to match the motion data and sensor data.
- “The model’s explanatory variables were the scaled sensor outputs and their derivatives.” – how did you calculate the derivatives?
- It is not clear how the ML models were trained. The details of the training data, validation data, and test data should be clarified. Moreover, the evaluation of the model performance should be from the test results rather than from the cross-validation. Also an intra- and inter-subject evaluation of the models should be discussed.
Reviewer 2 Report
Very well-written manuscript. The introduction states the motivation behind the work. Limitations of the present work are mentioned. Below are a few suggestions:
- Lines 122-128 mention the previous work done. Although it is explained how this work differs from reference # 55, the reviewer still wonders what this current work adds to the previous work.
- Line 76, there might be a "rotation" word missing after "internal/external".
- The authors state that 18 models were built and tested. However, it is not clear if all the models were random forest or any other algorithms were used.
- Figure 5 a and b shows box plots. It would be better if the authors clearly state what the red line in the box plot as well as the edges of the box represent. Also, it would facilitate the reading if the authors include the information regarding the actual and predicted lines in the legend (not only the caption) for Figure 5 c and d.
- What does 2 degree bin mean in the caption of Figure 6. More explanation is necessary.
- Lines 277-278 seem to be redundant.
- The major concern is that the authors do not give necessary information regarding the data processing for machine learning models. The reviewer suggests that the authors add a section where they clearly explain the data processing and its details. The reviewer wonders if the authors have used tested and validated the model. If so, what percentage of the data is used for testing and what percentage is used for validation. More information is required for this part.
- It is very beneficial that the authors address the limitations of the current work (lines 289-319). However, the first point is a little unclear. Why does removing the data from two people introduce interpolation to the system? Isn't it possible to simply remove the whole data generated with those experiments?
Reviewer 3 Report
- The author should highlight the significance of this work in the introduction in a better understandable way
- Besides the types of stain sensors mentioned in the introduction, there are many others, e.g., gel-based strain sensor. The authors are suggested to include some representative gel strain sensors (e.g., An Anti‐Freezing, Ambient‐Stable and Highly Stretchable Ionic Skin with Strong Surface Adhesion for Wearable Sensing and Soft Robotics." Advanced Functional Materials 31.42 (2021): 2104665),
- Figure 2 should be labeled clearly for better understanding.
- Could the authors show the drifting effect of current sensor? Reviewers are curious about how this outperformance other types of sensors.
Round 2
Reviewer 1 Report
The revised version is much better. However, the following points should be addressed:
- "All references to TKA incisions and patients, outside of a discussion of future applications and motivating background information, have been removed." -- I do not see this has happened.
- “Derivatives were calculated using finite differences between consecutive data points.” -- It would be good to show readers your sensor recordings and their derivatives graphically, say one set of data.
Author Response
- Thank you, we did make several changes during the first revision, and we have removed two additional references (Abstract, line 1; Method and Materials, line 91). The remaining references to TKA are all in the context of motivating context or discussion of anticipated future applications.
- Figure 5 has been added to convey an example of the sensor readings, as well as their derivatives.
- Additionally, we have conducted a thorough grammatical and editorial review of the manuscript that has hopefully improved the readability.
Reviewer 3 Report
N/A
Author Response
We have conducted a thorough editorial and grammatical revision that we feel has addressed the reviewer's request.